

# A framework for smartphone-enabled, patient-generated health data analysis

Shreya S. Gollamudi[1], Eric J. Topol[1,2,3] and Nathan E. Wineinger[1]

[1] Scripps Translational Science Institute, La Jolla, California, United States
[2] Department of Molecular and Experimental Medicine, The Scripps Research Institute, La Jolla, California, United States
[3] Division of Cardiovascular Diseases, Scripps Health, San Diego, California, United States

## ABSTRACT

**Background:** Digital medicine and smartphone-enabled health technologies provide a novel source of human health and human biology data. However, in part due to its intricacies, few methods have been established to analyze and interpret data in this domain. We previously conducted a six-month interventional trial examining the efficacy of a comprehensive smartphone-based health monitoring program for individuals with chronic disease. This included 38 individuals with hypertension who recorded 6,290 blood pressure readings over the trial.

**Methods:** In the present study, we provide a hypothesis testing framework for unstructured time series data, typical of patient-generated mobile device data. We used a mixed model approach for unequally spaced repeated measures using autoregressive and generalized autoregressive models, and applied this to the blood pressure data generated in this trial.

**Results:** We were able to detect, roughly, a 2 mmHg decrease in both systolic and diastolic blood pressure over the course of the trial despite considerable intra- and inter-individual variation. Furthermore, by supplementing this finding by using a sequential analysis approach, we observed this result over three months prior to the official study end—highlighting the effectiveness of leveraging the digital nature of this data source to form timely conclusions.

**Conclusions:** Health data generated through the use of smartphones and other mobile devices allow individuals the opportunity to make informed health decisions, and provide researchers the opportunity to address innovative health and biology questions. The hypothesis testing framework we present can be applied in future studies utilizing digital medicine technology or implemented in the technology itself to support the quantified self.

## INTRODUCTION

Empowered patients and health care consumers (*Topol, 2015*) have aligned with health data tracking technologies to create the quantified self movement (*Swan, 2013*). Quantified self involves the use of tracking one's own health-related data to understand trends and potentially alter behavior in order to achieve a health goal. The size and

Corresponding author
Nathan E. Wineinger,
wineinger.nathan@scrippshealth.org

scope of health tracking has and continues to expand with the advent of digital medicine and mobile health (mHealth) technologies enabled by smartphones and connected device infrastructures (*Steinhubl, Muse & Topol, 2015*). For example, in certain individuals traditional daily weight monitoring has been supplemented by apps that track food intake and devices that monitor physical activity in order to achieve this health goal—a part of the 58% of mobile phone users that have downloaded a health-related mobile app (*Krebs & Duncan, 2015*). Researchers have shown that in some cases interventions using this technology alone can improve health outcomes, though this result is far from universal (see *Free et al. (2013)* for review) with the disparity likely due to numerous factors including poor adherence and fatigue (*Shaw et al., 2016*), and failure of the intervention to change behavior (*Patel, Asch & Volpp, 2015*).

While new, better, more user-friendly technologies will continue to be developed, vying for the appreciable forecasted growth of the industry (*Statista, 2016*), we and others believe the future of the field is not simply in the devices and software themselves, but in the data they generate (*Gibbs, 2015*). Such data can help guide individual health decisions—the crux of the quantified self movement—but can also be used to address novel human health and biology questions (*Steinhubl et al., 2015*). Yet there exists a sizable gap between the data that is generated and the methods available to analyze and interpreted such data (*Fawcett, 2015*). Indeed, most such devices and apps simply display the data, leaving any inference up to the user and anyone the user wishes to share the data with (e.g., their physician). However, even the most data experienced users and health care providers may find identifying subtle trends in such complex data a daunting task. These challenges also extend to researchers who may wish to examine data captured from these technologies as, in addition to inherent technical obstacles such as data collection and data security, few analytic methods are established and general software packages are not readily available.

These technologies present the opportunity to examine health data in nontraditional ways. Rather than large intermittent gaps in health measures between doctor or study visits, data can be collected in relatively high resolution. Such high resolution data allows users and researchers to detect unique trends and relationships that were never before possible. For example, a diabetic patient using a continuous glucose monitor can now assess their minute-to-minute health rather than rely on single, low resolution measures such as hemoglobin A1c levels, which does not accurately assess health variability (*Virtue et al., 2004*). Furthermore, access to such high-resolution human health data in nontraditional settings (e.g., normal, at home environment) allows us to evaluate "real world" health and not be relegated to artificial worlds created in clinical trials that suffer from poor clinical practice adoption (*Goss, Elmore & Lessler, 2003*). Yet in order to form scientific conclusions in this new frontier, novel methods and adaptations of existing approaches must be developed to account for the intricacies of patient-generated data.

While numerous digital medicine biosensors, devices, and applications have been manufactured to measure various aspects of human physiology and exposome, perhaps

no metric epitomizes both the contemporary challenge and opportunity of this field more than blood pressure. Heart disease is the leading cause of death in the United States, with hypertension being the leading contributor of disease (*National Center for Health Statistics, 2012*). With proper management, hypertension can be controlled and the health consequences of uncontrolled hypertension can largely be avoided. Nevertheless, in the United States only 48% of individuals with hypertension have their condition under control (*Farley et al., 2010*). It is unsurprising that hypertension has been a target for patient-centric, mHealth disease management (*Logan, 2013*). Yet the technology to continuously monitor blood pressure in the outpatient setting is still developing as manufacturers have not entirely solved the technical aspects of truly passive monitoring. Thus, the current state of the field largely includes mobile blood pressure cuffs in which readings are initiated by the user. As users may take a reading at any time, evaluating temporal trends in this data requires added consideration.

In this study, we present a hypothesis testing framework in which we examined blood pressure readings taken at variable, uncontrolled time points in individuals enrolled in a smartphone-based health monitoring intervention trial—though the approach we present can be adapted to other similarly structured data. In total, 38 study participants recorded and provided us with blood pressure data on 6,290 occasions. We find that by leveraging all data across individuals we were able to detect an approximately 2 mmHg decrease in blood pressure over a six month trial, despite considerable intra- and inter-individual variation. We then discuss how this and other techniques can be implemented in data analyses of the quantified self and in future study designs.

## METHODS

### Study participants

The present investigation is a sub-analysis of a study conducted by the Scripps Translational Science Institute named the Wired for Health (WFH) study (*Bloss et al., 2016*). In brief, the WFH study was a six month, randomized-controlled trial investigating the practice of a smartphone-based health monitoring program in individuals with chronic disease, and was accompanied by an online and mobile tracking infrastructure. Eligible participants were over the age of 18 who were insured by Scripps Health and had submitted at least one health insurance claim for hypertension, diabetes, or cardiac arrhythmia in 2012. Participants were equally randomized to the control or monitoring arms (details below). In total, 160 participants enrolled in the WFH study, with the majority (51.3%) being in the top quartile of health insurance claims in 2012. Only individuals with hypertension (n = 135) were considered in the present study. Among those, 112 completed the study, including 53 in the control and 59 in the monitoring group. After screening device readings data for technical limitations (n = 19) and study noncompliance (n = 2), 38 hypertensive individuals from the monitor group had complete readings data. This study focuses on these 38 individuals. This study was approved by the Scripps Institutional Review Board (IRB-12-6019), and all study participants provided informed consent.

## Monitoring intervention arm

Hypertensive study participants in the monitoring group were provided with comprehensive mobile blood pressure monitoring system: a Withings Blood Pressure Monitor, an iPhone 4 or 4s with linked applications, iPhone applications, and an online and mobile HealthyCircles account. HealthyCircles is a Qualcomm Life health care coordination and management platform with an integrated suite of management and consumer portals that can deliver chronic disease education and connect users to their families, caregivers, and health care professionals. Individuals were instructed to measure their blood pressure using this system twice a day, three days a week, with the first measurement in the morning. If participant measurements dropped below a desired level of compliance (less than three measurements a week for two consecutive weeks) the participant was sent an email through their HealthyCircles account reiterating the measurement schedule. Participants were also encouraged to take extra measurements if deemed appropriate. Device readings data was collected using Qualcomm Life's cloud-to-cloud data integration capability.

## Variables of interest

The primary outcome of the present study was device-collected blood pressure measurements in the 38 individuals with complete readings data in the monitoring group. The primary independent variable is time since the beginning of enrollment in the study. The hour during the day the measurement was taken was considered as a covariate.

## Statistical analyses

Device readings data was analyzed using two approaches: (1) a multiple N-of-1 approach in which the data from each study participant was analyzed individually; and (2) a mixed model approach combining all individuals for analysis. Specific details on each model are available in Appendix S1.

In the multiple N-of-1 approach, blood pressure measures were regressed on time enrolled in the study using linear regression, accounting for the time of day at which the measurements were taken. Alternative covariance structures were modeled, but results were consistent with those obtained from simple linear regression. In all cases, the effect of blood pressure over time (i.e. slope) was recorded. Slope averages, inverse variance weighted averages, and bootstrapped confidence intervals were calculated.

Alternatively, repeated measures mixed models were constructed to assess blood pressure over time across all individuals. The general structure of the model is:

$$Y = X\beta + Zu + e$$

where $Y$ is the vector of blood pressure measures, $X$ is a matrix of independent variables (i.e. intercept, time, and covariates) with fixed effects $\beta$, $Z$ is a matrix indicating the structure of the between subject random effects $u$ with covariance matrix $G$, and $e$ is the random error with covariance matrix $R$. It follows that $Var(Y|u) = R$ and $Var(Y) = V = ZGZ^T + R$.

In the present study, $R$ represents the potential time-dependency of measures within subjects. It follows that $R$ and subsequently $V$ are block diagonal. Below we refer to the

block diagonal elements of **V** as $\Sigma$, and the elements of **R** as $\Sigma_R$. The primary hypothesis tested was H$_0$: $\beta_{\text{time}} = 0$. That it, was there a linear change in blood pressure measures in the study population after accounting for individual variation and potential time-dependency between measures? Time of day (in hours) was modeled as a fixed effect covariate with 24 levels. We do note that that random effects component of this model indirectly accounts for static individual-specific covariates (e.g., baseline BMI). Thus, no other covariates were modeled.

Three distinct covariance structures were modeled that were appropriate for the source of the data and minimally complex on account of the large dimensions of **V**: (1) compound symmetric structure $\Sigma$ (i.e. random effects only); (2) a first-order autoregressive structure with random effects; and (3) a spatial power law/generalized autoregressive structure with random effects. Model fit was assessed using AIC and BIC.

Finally, a sequential analysis approach was implemented using the mixed model approach described above. Study device readings data was collected over a roughly ten and one-half month period from the middle of August 2013–July 2014. Data was partitioned into eleven cumulative monthly periods. For example, one data partition included all device readings data from the first month of the study, another included device readings data from the first two months of the study, and so on. A mixed model with spatial power law covariance structure with random effects was applied to each of the data partitions. Additionally, a sequential analysis approach was implemented in an N-of-1 framework where readings from each individual alone were partitioned into monthly blocks similar to that described above. Among individuals who did demonstrate a significant increase or decrease in blood pressure at six months, the goal was determine if changes in blood pressure could be observed prior to the conclusion of the individual's study participation (e.g., if an individuals demonstrated a decrease in blood pressure over six months, would we have observed that result after five months). Likewise, a spatial power law covariance structure was assumed.

## RESULTS

Demographic information on study participants is presented in Table S1. Among the 59 original study participants assigned to the monitoring intervention arm that completed the study, 38 had complete device readings data. This cohort was predominantly Caucasian (87%) and largely female (74%) with an average age of 57. A number of participants did not own a smartphone prior to enrollment in the study (21%) with half owning an iPhone. Responses to health-related survey questions are presented in Table S2. We saw a general increase in overall health by the end of the study. There was notable decrease in smoking frequency and increase in exercise frequency. However, we note that these health improvements were also observed in the control arm of the original study (data not presented), suggesting that the monitoring intervention itself had no discernible impact on these traits.

The implied goal of individuals participating in the study is better management of their condition. In this regard, we considered observed blood pressure device readings

collected over the course of the study as outcomes of interest. Single blood pressure readings taken at, for example, the enrollment and end of study visit can provide some level of inference on blood pressure changes. However, this approach ignores all data that could be generated between these time points and is vulnerable to biases and natural variation. Rather, we feel approaches which leverage the entirety of the data are preferential. By utilizing Qualcomm's cloud-to-cloud data integration capability, we were able to capture the measure and time recorded of each blood pressure reading on 38 individuals in the monitoring group. While our inference is based on these 38 individuals and is limited, we present our mathematical framework and modeling below in attempt to answer the seemingly simple question: was blood pressure changing over the course of the study?

In total, we collected 6,290 systolic and 6,265 diastolic blood pressure readings from these individuals (Data S1 and S2). Device readings were recorded roughly uniform over the course of the study (Fig. S1). The number of readings taken varied between individuals, with an average of 165 readings per person (sd = 70, min = 61, max = 416). The time of day that readings were taken was also variable, with a large proportion of measurements taken in the early morning and afternoon. Few readings were taken in the late morning and at night, though this was not necessarily surprising given we asked participants to use the device in the morning, presumably before day time activities (Fig. S2).

### Multiple N-of-1 approach

We first assessed the effect of time on blood pressure on each study participant individually (Figs. 1 and S3). Among the 38 participants, 18 had nominally statistically significant ($p < 0.05$) changes in systolic blood pressure and 21 had significant changes in diastolic blood pressure. However, the number of participants with a significant decrease in systolic blood pressure (n = 9) was equal to the number with an increase (n = 9), and was similarly true for diastolic blood pressure (decrease: n = 12, increase: n = 7; $p = 0.36$). This result also held when we examined the estimated effects from all participants, regardless of p-value. There were 20 individuals with a decrease (i.e. negative slope) in systolic blood pressure against 18 with an increase ($p = 0.87$), and 21 with a decrease and 17 with an increase in diastolic blood pressure ($p = 0.62$). In efforts to summarize these results across individuals, we calculated the mean slope and mean slope weighted by the square root of the number of readings each participant recorded. We found no evidence towards an overall decrease in systolic or diastolic blood pressure. The weighted mean change in systolic blood pressure was −1.7 mmHg (95% CI: −4.7, 1.4) and diastolic blood pressure was −1.9 mmHg (95% CI: −4.0, 0.1). Results from the unweighted calculations were similar.

### Mixed model approach

We then pooled device readings data from all study participants, and assessed the effect of time on blood pressure using the mixed model framework described previously with three possible covariance structures of **R**, where $\Sigma_R$ was: (1) compound symmetric;

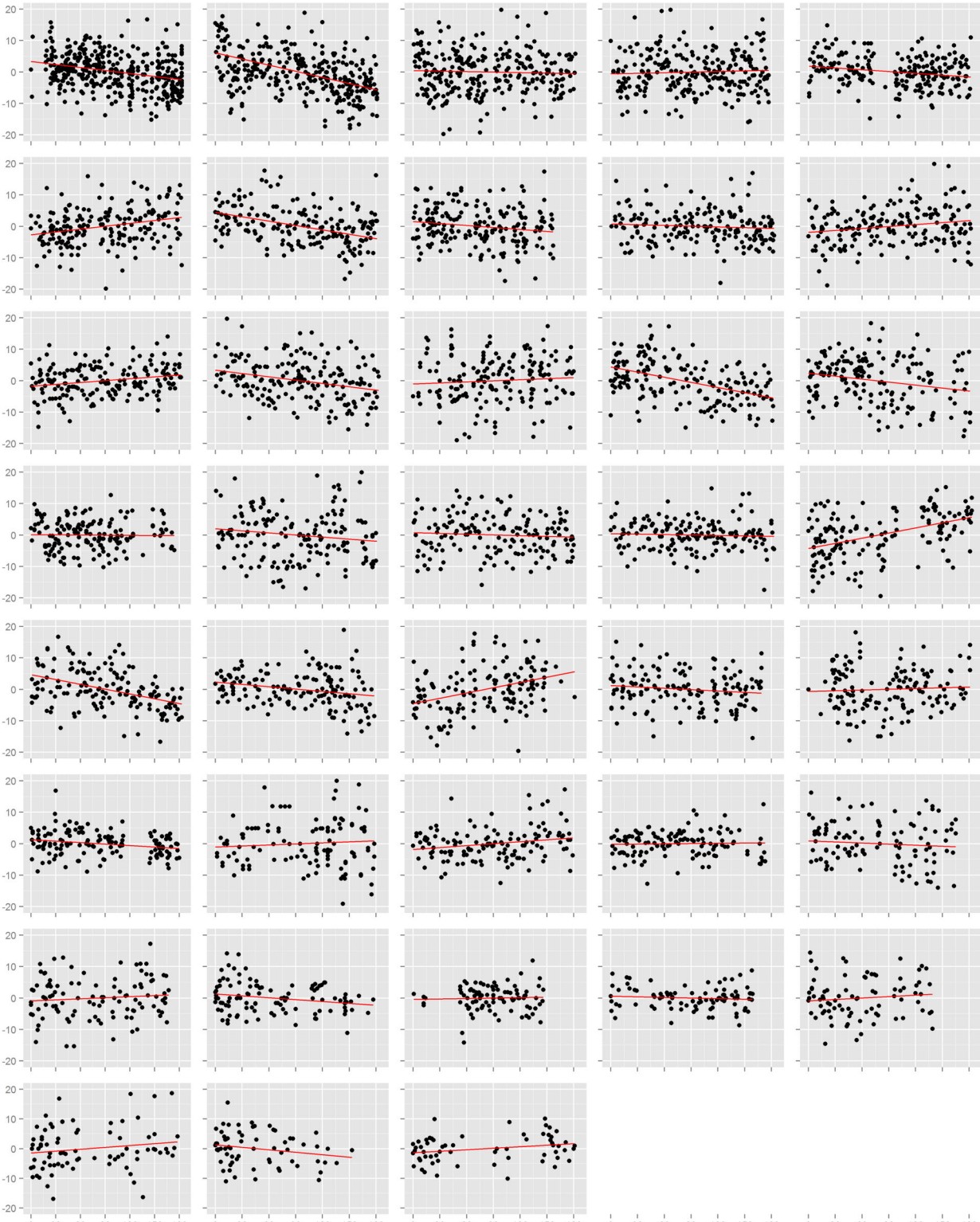

**Figure 1 Normalized diastolic blood pressure readings.** Each box is one study individual. Points are arranged along the x-axis which represents the time in days from the beginning of the study, and along the y-axis which represents the normalized diastolic blood pressure reading recorded at that time. The red line is the least squares regression line. Individuals are ordered left to right, top to bottom according to the number of readings recorded.

(2) first-order autoregressive; or (3) generalized autoregressive for unequally spaced data (i.e. spatial power law).

We encountered one issue when applying the spatial power law structure mixed model. As opposed to the first-order autoregressive model in which

$$\sum\nolimits_R = \sigma^2 \begin{pmatrix} 1 & \rho & \rho^2 & \\ \rho & 1 & \rho & \cdots \\ \rho^2 & \rho & 1 & \\ & \vdots & & \ddots \end{pmatrix}$$

where $\rho$ is the correlation between two successive measures on the same subject, the spatial power law structure is a generalized form of the first-order autoregressive model in which

$$\sum\nolimits_R = \sigma^2 \begin{pmatrix} 1 & \rho^{t_2-t_1} & \rho^{t_3-t_1} & \\ \rho^{t_2-t_1} & 1 & \rho^{t_3-t_2} & \cdots \\ \rho^{t_3-t_1} & \rho^{t_3-t_2} & 1 & \\ & \vdots & & \ddots \end{pmatrix}$$

where $t_k$ is the time of the $k^{\text{th}}$ measurement, and the difference $t_i - t_j$ is the lag between the $i^{\text{th}}$ and $j^{\text{th}}$ measure. When measurements are equally spaced, the spatial power law models simplifies to a first-order autoregressive model as the lag between measures is constant. Though we instructed study participants to measure their blood pressure at certain intervals, the time individuals chose to take these readings, and thus the lag between measures, varied considerably. In this context, we expected the spatial power law structure to be appropriate for the data collected. However, while the majority of consecutive measures had lags of six hours or more (Fig. 2), we discovered a small number of consecutive measures with short lag that led to issue we alluded to above. To demonstrate this issue between consecutive measurements with short lag, consider the following example: three measurements are recorded at times labeled $t_1$, $t_2$, and $t_3$. In this case,

$$\sum\nolimits_R = \sigma^2 \begin{pmatrix} 1 & \rho^{t_2-t_1} & \rho^{t_3-t_1} \\ \rho^{t_2-t_1} & 1 & \rho^{t_3-t_2} \\ \rho^{t_3-t_1} & \rho^{t_3-t_2} & 1 \end{pmatrix}$$

However, when two measurements are taken relatively close together, say at $t_1$ and $t_2$ compared to $t_3$, then $t_2 - t_1$ is relatively close to zero and $t_3 - t_1 \approx t_3 - t_2$. The result on $\Sigma_R$ is:

$$\sum\nolimits_R \approx \sigma^2 \begin{pmatrix} 1 & 1 & \rho^{t_3-t_1} \\ 1 & 1 & \rho^{t_3-t_1} \\ \rho^{t_3-t_1} & \rho^{t_3-t_2} & 1 \end{pmatrix}$$

As can be seen, the first and second columns and rows are roughly equivalent, leading to singularity in this matrix, non-convergence, and inestimable effects. It should be noted that this issue does not manifest in the compound symmetric and first-order autoregressive models.

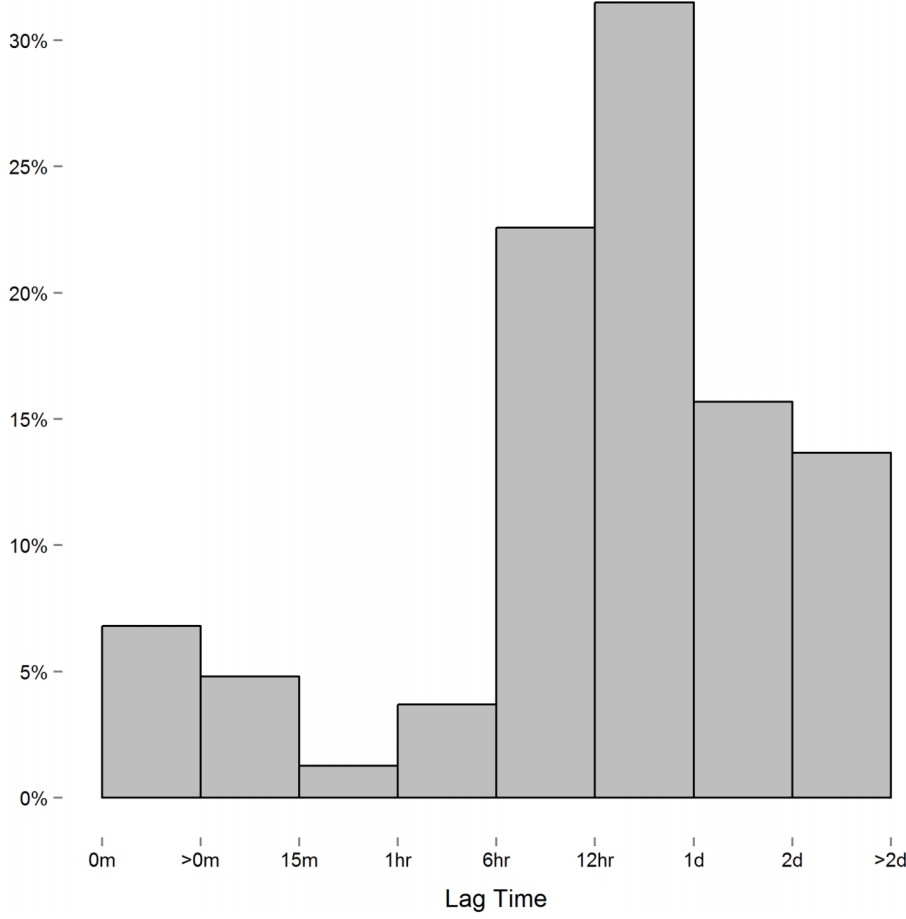

**Figure 2 Histogram of the lag between consecutive measures.** Measures recorded near each other relative to others can lead to singularity in $\Sigma_R$.

In order to solve this issue, we merged data readings taken within short periods of each other by taking the average over that time. We found that merging readings within a one hour period of each other eliminated the singularity of $\Sigma_R$ while minimizing the number of readings manipulated. We still ran into singularity issues when using shorter time intervals (e.g., 15 and 30 min minimums). In total, we merged 826 systolic and 801 diastolic blood pressure readings within an hour of each other, with most merging (61%) being two readings recorded within an hour. The subsequent dataset resulted in 5,464 systolic and diastolic readings.

Mixed model results from the one-hour merged dataset are presented in Table 1. For each model assessed, there was an approximately 2 mmHg decrease in systolic and diastolic blood pressure across the sample over the study period ($p < 0.001$ in all cases). While both the first-order autoregressive and spatial power law models with random effects had better model fit than the random effects model alone, the first-order autoregressive model had slightly better model fit (the estimate of $\rho$ was near 0.2 regardless of the approach). It should be noted that when we tested various merging strategies to eliminate the singularity in the spatial power law $\Sigma$ matrix (e.g., averaging blood pressure across the

**Table 1 Mixed model results.**

|  | Σ | AIC | BIC | Estimate (CI) | p |
|---|---|---|---|---|---|
| Systolic | RE | 41,604 | 41,608 | −2.11 (−3.13, −1.09) | $5.19 \times 10^{-5}$ |
|  | RE + AR(1) | 41,495 | 41,500 | −2.11 (−3.29, −0.93) | $4.45 \times 10^{-4}$ |
|  | RE + SP | 41,546 | 41,550 | −2.04 (−3.11, −0.98) | $1.80 \times 10^{-4}$ |
| Diastolic | RE | 36,725 | 36,729 | −2.04 (−2.69, −1.39) | $9.41 \times 10^{-10}$ |
|  | RE + AR(1) | 36,620 | 36,624 | −2.06 (−2.81, −1.31) | $8.17 \times 10^{-8}$ |
|  | RE + SP | 36,705 | 36,710 | −2.05 (−2.72, −1.37) | $2.83 \times 10^{-9}$ |

**Note:**
RE, random effects only (compound symmetric $\Sigma_R$); RE + AR(1), first-order autoregressive model with random effects; RE + SP, spatial power law with random effects.

entire day) sometimes the spatial power law model had the better fit. This suggests that while accounting for time dependencies was important, the precise structure appeared to be less so. This was true even though the spatial power law model appeared to be more appropriate, given the readings were generally unequally spaced.

*Sequential analysis approach*

We next implemented a sequential analysis approach in which we modeled a spatial power law structure with random effects on sequential subsets of the one-hour merged datasets. The goal of this approach was to determine if and when we could have arrived at our primary conclusion (i.e. systolic and diastolic blood pressure decrease 2 mmHg over the study) prior to the end of the study. While the device readings data across all study participants was collected over ten and a half months, after the first seven months of the study we would have arrived at a similar conclusion (Figs. 3 and S4). Though all 38 participants had completed the study or were enrolled by that time, and 4,686 blood pressure measures (86%) had taken place by then, the prospect of arriving at a conclusion prior to the designed end of the study has some benefit—particularly so when data is collected using digital medicine devices as we discuss below. Furthermore, when applying a sequential analysis approach in an N-of-1 framework, we were able to determine that 13 of the 15 individuals with nominally significant systolic blood pressure changes, and 14 of the 18 individuals with diastolic blood pressure changes over six months demonstrated this trend ($p < 0.05$) by the fifth month of their study enrollment.

# DISCUSSION

As digital, smartphone-enabled, and other patient-centric medical and health technologies have the potential to improve individual health and the overall health care system, the quest to develop the "best" technologies will remain ongoing. However, we and others feel the future of this field is not simply in the devices, sensors, software, and wearables per se (*Gibbs, 2015*), but in what the data generated from these tools can tell us about human health and biology. Above, we present a framework for hypothesis testing on unequally spaced time series data—a common feature of data generated from these technologies. Applied to a subset of hypertensive individuals enrolled in an interventional trial, we discovered that individuals participating had, on the whole, a roughly 2 mmHg decrease of systolic and diastolic blood pressure over a six month

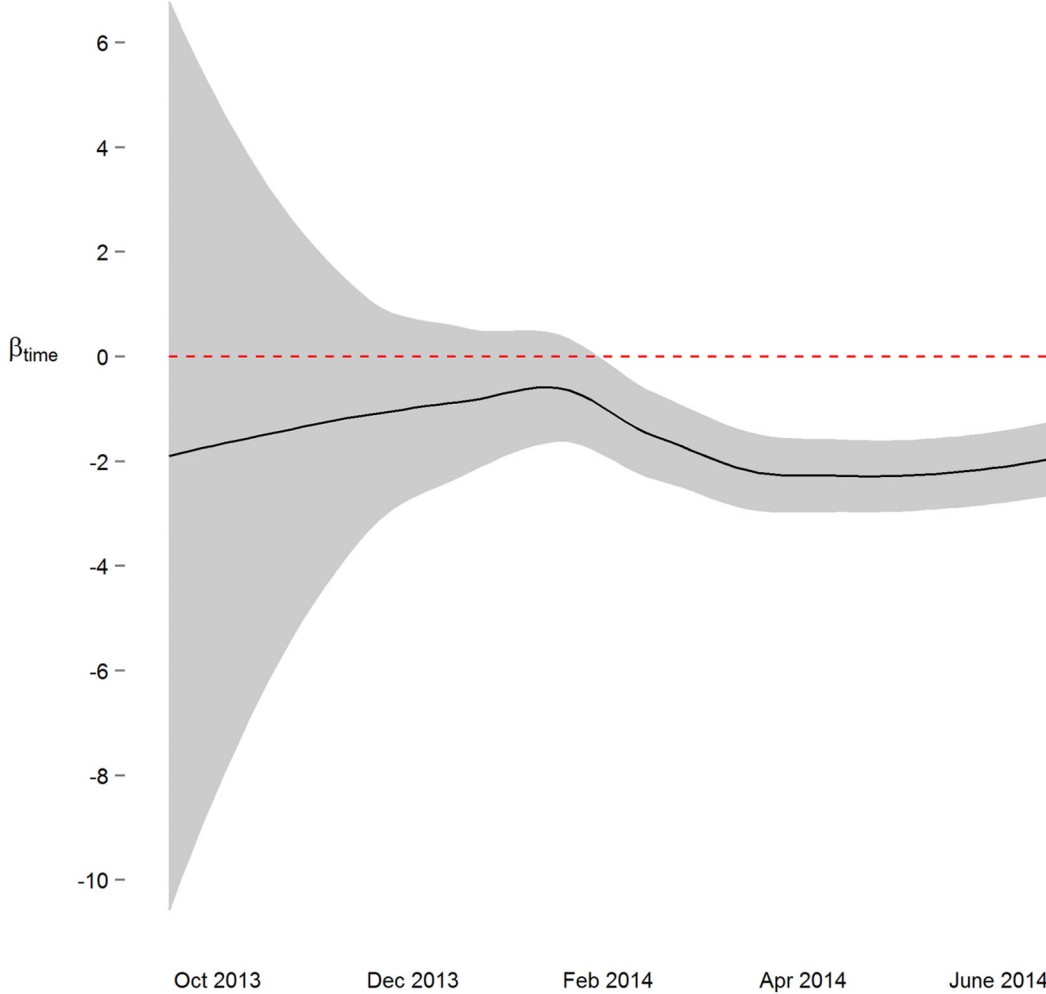

**Figure 3 Parameter estimate and corresponding 95% confidence interval assessing change in diastolic blood pressure over the course of the study.** By March 2014, three months prior to the conclusion of the study, the primary study outcome (roughly 2 mmHg decrease) was observable.

period. Using the methods presented we were able to observe this distinction despite considerable intra- and inter-individual variation in blood pressure measures, and without a rigorously structured readings schedule.

The context of this discovery should, however, be considered. As stated above, we detected a statistical increase in systolic (n = 9) and diastolic blood pressure (n = 7) in some study participants, while device readings were (unsurprisingly) quite variable even among study participants that clearly showed a trend towards lowering blood pressure. We have recently demonstrated some explanation for this heterogeneity in study outcomes (*Kim, Wineinger & Steinhubl, 2016*). But this observation demonstrates the advantage, yet caution, needed when performing analyses and interrupting results that leverage large amounts of high-dimensional patient-generated data. On one hand, this presents an opportunity to identify subtle trends in health data that might otherwise be difficult to observe—such as single measurements intermittently collected during a physician visit. On the other hand, large amounts of data can make it possible to detect
any trend even if this trend does not substantially contribute to our knowledge of human health and biology. For these reasons we find it more informative to reflect on the magnitude of the effect and less on the magnitude of the p-value. Complementary approaches which specify the 'null' model as a minimally meaningful effect (i.e., non-zero contrast matrix) may be more appropriate in such cases.

Importantly, while we focused on time enrolled in the study as the independent variable of interest, the framework we present can also be adapted to examine other temporal relationships in similarly generated data. For example, this framework can be used to compare data captured between discrete conditions such as intervention versus control, or to differentiate based on other quantitative measures. In these cases, the $X$ matrix presented above can be remodeled to reflect the desired design matrix. Additionally, while our mixed model approach modeled data across all 38 study participants collectively, this framework can be also adapted to examine temporal trends on a single individual, such as that of an N-of-1 crossover design or how we examined individual trends in the N-of-1 sequential analysis approach above. In this case, the $Z$ matrix can be omitted on account of no between subject effects, while $R$ would remain.

One of the more intriguing aspects of this technology as a tool to enhance individual health is that data is collected, stored, and presented digitally without the need for direct interaction between the user and (as traditional) health professional. Likewise, we feel the processes of data inference can also be built into the technology to bypass the need for data interpretation by a professional data analyst. Certainly many technologies have such tools. Yet as new methods and extension of existing approaches, such as the framework we presented, are developed, these will need to be implemented into the technology in order to provide users with the best opportunity to make informed health decisions based on this data. The most immediate way this can be accomplished is by coding these methods directly on the device or software, or accessible to a cloud server where computations can be performed. However, other options include crowd-sourced initiatives such as an app store, where the public can design specialized software which provides automated analyses and interpretation of data back to the user.

Alternatively, the digital nature of data obtained from this technology opens up a number of interesting possibilities for researchers. Again, because data can be continuously collected without the need for personnel (e.g., study coordinators) to interact with users/study participants, approaches which benefit from data analyses over the course of the study may prove beneficial. We attempted to show this in the sequential analysis approaches above in which we were able to arrive at the primary study result over three months prior to the end of the study. Methods like this can be implemented directly into the study design, such as those in adaptive clinical trials. Moreover, because data on individuals can be recorded, collected, analyzed, and interpreted in real time, concepts such as early stoppage due to success or futility can apply not only to the study itself, but study participants as well—thereby minimizing risks, reducing costs, and forming conclusions earlier.

Data collected from digital medicine and smartphone-enabled health technologies offers tremendous potential to learn more about human health and biology. We applaud

manufacturers for striving towards more comprehensive monitoring technologies, and when applicable encourage researchers to use this data source to help address research questions of interest.

## ACKNOWLEDGEMENTS

Steven R. Steinhubl, M.D. provided study suggestions; Qualcomm Life made device readings data available.

### Funding
This research is funded in part by a NIH/NCATS flagship Clinical and Translational Science Award Grant (1UL1 TR001114), Qualcomm Foundation Scripps Health Digital Medicine Research Grant, and Scripps Health's Division of Innovation and Human Capital and Division of Scripps Genomic Medicine. Support for the study is also provided by HealthComp Third Party Administrator, Sanofi, AliveCor, and Accenture. The funders had no role in study design, data collection and analysis, decision to publish, or preparation of the manuscript.

### Grant Disclosures
The following grant information was disclosed by the authors:
NIH/NCATS flagship Clinical and Translational Science: 1UL1 TR001114.
Qualcomm Foundation Scripps Health Digital Medicine Research.

### Competing Interests
The authors declare that they have no competing interests.

### Author Contributions
- Shreya S. Gollamudi analyzed the data, wrote the paper, prepared figures and/or tables, reviewed drafts of the paper.
- Eric J. Topol conceived and designed the experiments, reviewed drafts of the paper.
- Nathan E. Wineinger conceived and designed the experiments, analyzed the data, wrote the paper, prepared figures and/or tables, reviewed drafts of the paper.

### Clinical Trials
The study was registered at https://clinicaltrials.gov (NCT01975428).

### Human Ethics
The following information was supplied relating to ethical approvals (i.e., approving body and any reference numbers):
This study was approved by the Scripps Institutional Review Board, and all study participants provided informed consent. IRB-12-6019.

## Data Deposition

The raw data has been supplied as Supplemental Dataset Files.

## Supplemental Information

Supplemental information for this article can be found online at http://dx.doi.org/
10.7717/peerj.2284#supplemental-information.

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
