# Peer review of "A framework for smartphone-enabled, patient-generated health data analysis"

_PeerJ, doi:10.7717/peerj.2284_

## Round 0.1 · original submission · Major Revisions

Thank you for the very interesting article. I have received 2 expert reviews and in my opinion their concerns are entirely addressable. I encourage you to resubmit with a rebuttal.

George

·

Basic reporting

No Comments

Experimental design

The paper makes the claim that the hypothesis testing framework provides can be applied to further studies.

However, the formal model definition is unclear for both the N-of-1 model and the Mixed Effects Model.

The N-of-1 model is described a linear regression model, but it is unclear in the paper what the covariate are in the model. It is stated that model accounted for the time of day, but the details of the approach are not clarified. For example: Was the time of day modeled using fixed effects for each hour/six-hours or was some other approach used? Were the systolic/dialostic measurements modeled individually and were they modeled using the same covariates?

The mixed effects model was formally specified, but the details are unclear. For example, are the fixed effects modeled as regression type slopes, or fixed changes in means. I understand the auto-regressive structure and it is clear why the power law approach was used, but what are the fixed effects? Additionally, though this outside my expertise and is therefore a general question, is there daily seasonality and if so was this modeled or addressed in any way other than the autoregressive structure?

An appendix that explains the model details and shows how to recreate the model in other studies would address many of these concerns.

Finally, just a pure comment but was there any consideration of modeling the systolic and diastolic measurements using a multivariate model and incorporating the correlation between these two measures?

Validity of the findings

If I understand the stated results correctly the N-of-1 model is a null result at .05 alpha level (your stated CI's overlap 0, though only barely in the case of diastolic). Statistically speaking this is a null result and even the statement 'limited evidence' is too strong a claim.

Also, I don't have any sense of the general quality of fit metrics for the mixed effects model. You show that the AIC and BIC metrics are better with the model, but it is unclear what the null model you are comparing against is and how much the AIC actually improves. Looking at Figures 1 and 3 I am concerned that you have are detecting a small signal in a very large amount of noise. You have enough degrees of freedom to detect small differences (it is unclear because the model is not formally stated, but I would suspect you have at least 1,000 df if not several times that) so the low p-values may just reflect a very high level of sensitivity.

If I understand Supplemental Figures 1 and 3 correctly, some individuals have increasing slopes, others have decreasing slopes and the noise is often quite large compared to the slope (suggesting you would have low R^2 values) and it is unclear if the model diagnostics are solid.

These issues are purely with the statistical conclusions, but I would like to see some additional diagnostics that help put the results into context. For example, have the auto-correlation or spatial power law reduced the assumed temporal correlation, and do the residuals truly look like noise? Are there any signs of model fitting problems, or is it merely the case that the effect size is very small relative to the noise but the effect really is real?

Finally, the changes may in fact be negative for the population, but a large percentage of the population would be estimated to have positive changes, so do these effects have any practical significance?

Additional comments

It feels like you really have two papers, and this is where I think the paper has the biggest challenge.

The way the paper is described you are going to provide a useful framework for modeling and then are going to show an example of how this can be used. What happens instead is that you have a paper where you describe an experiment, then you describe the model, then you state the results of the experiment while explaining some of the modeling challenges. Is this a paper focused on the methodology or the experiment? If the focus is on the methodology (which is what it feels like from the explanation) then more details need to be provided to flesh out what makes this model particularly powerful. If the focus is on the experimental results then the title is misleading.

If I understand the point of the paper (and this may be a point of confusion on my side) then you want to show how to model unstructured time series data where you can't use ARIMA or exponential smoothing or other sophisticated time series models. So you model these effects using a mixed effects model that captures all these features.

But the details are very unclear. What are your fixed effects, what random effects did you actually use. There is no specification of the model at a detailed level and is unfortunate because I found it unclear even understanding what the benefit is. This may be a case that you are so familiar with what you are doing that the lack of clarity is not obvious. In my case it may be because my expertise is primarily in structured time series, traditional statistics, and some biostatistics and the audience may be expected to be more comfortable with these ideas. But for me, looking at it from a modeling perspective, I didn't really get a proper sense of the model.

·

Basic reporting

no comment

Experimental design

No comments

Validity of the findings

Valid but non-generalizable as the sample were primarily middle age female.

Additional comments

This manuscript concerns a timely topic and is well written with a pertinent introduction concerning the interpretation of quantified health information coming from the growing field of personal passive measurement devices.

The use of regression, mixed model regression, and sequential analysis is appropriate and prudent.
The authors stated that no covariates where introduced in the model. This is curious to me. Shouldn’t BMI and or medication use be considered as confounders? The authors should address this issue.

In all the manuscripts represents a much needed advance in the field of interpreting data from the ever growing device market.

---

## Round 0.2 · accepted · Accept

I have examined the revision. Thank you for throughly addressing the concerns raised and supporting PeerJ